# Respiratory Rate Extraction from Neonatal Near-Infrared Spectroscopy Signals

**DOI:** 10.3390/s23094487

**Published:** 2023-05-05

**Authors:** Naser Hakimi, Mohammad Shahbakhti, Jörn M. Horschig, Thomas Alderliesten, Frank Van Bel, Willy N. J. M. Colier, Jeroen Dudink

**Affiliations:** 1Artinis Medical Systems, B.V., Einsteinweg 17, 6662 PW Elst, The Netherlands; 2Department of Neonatology, Wilhelmina Children’s Hospital, University Medical Center Utrecht, Lundlaan 6, 3584 EA Utrecht, The Netherlands

**Keywords:** near-infrared spectroscopy, neonates, respiratory rate, cerebral oxygenation, signal quality assessment

## Abstract

*Background:* Near-infrared spectroscopy (NIRS) relative concentration signals contain ‘noise’ from physiological processes such as respiration and heart rate. Simultaneous assessment of NIRS and respiratory rate (RR) using a single sensor would facilitate a perfectly time-synced assessment of (cerebral) physiology. Our aim was to extract respiratory rate from cerebral NIRS intensity signals in neonates admitted to a neonatal intensive care unit (NICU). *Methods:* A novel algorithm, NRR (NIRS RR), is developed for extracting RR from NIRS signals recorded from critically ill neonates. In total, 19 measurements were recorded from ten neonates admitted to the NICU with a gestational age and birth weight of 38 ± 5 weeks and 3092 ± 990 g, respectively. We synchronously recorded NIRS and reference RR signals sampled at 100 Hz and 0.5 Hz, respectively. The performance of the NRR algorithm is assessed in terms of the agreement and linear correlation between the reference and extracted RRs, and it is compared statistically with that of two existing methods. *Results:* The NRR algorithm showed a mean error of 1.1 breaths per minute (BPM), a root mean square error of 3.8 BPM, and Bland–Altman limits of agreement of 6.7 BPM averaged over all measurements. In addition, a linear correlation of 84.5% (*p* < 0.01) was achieved between the reference and extracted RRs. The statistical analyses confirmed the significant (*p* < 0.05) outperformance of the NRR algorithm with respect to the existing methods. *Conclusions:* We showed the possibility of extracting RR from neonatal NIRS in an intensive care environment, which showed high correspondence with the reference RR recorded. Adding the NRR algorithm to a NIRS system provides the opportunity to record synchronously different physiological sources of information about cerebral perfusion and respiration by a single monitoring system. This allows for a concurrent integrated analysis of the impact of breathing (including apnea) on cerebral hemodynamics.

## 1. Introduction

Monitoring changes in the respiratory rate (RR) of term and preterm infants admitted to the neonatal intensive care unit (NICU) plays a vital role in the timely detection of abnormal respiratory events such as tachypnea and apnea [1,2,3,4,5]. According to the literature, almost half of the neonates in the NICU were admitted due to respiratory morbidity [6]. Respiratory events and associated desaturation events affect cerebral physiology and contain relevant information about cerebral hemodynamics [7,8,9,10]. Hence, continuous monitoring of RR changes in hospitalized neonates and the impact on cerebral hemodynamics is of importance in clinical practice [11,12,13].

For monitoring cerebral hemodynamics, near-infrared spectroscopy (NIRS) can be employed, which is a non-invasive optical neuroimaging modality for measuring oxygenated (O2Hb) and deoxygenated (HHb) hemoglobin concentrations associated with neural activities in the cerebral cortex [14]. NIRS is becoming part of routine clinical practice in an increasing number of NICUs worldwide [15,16]. Aside from cerebral tissue oxygen saturation measured with NIRS, the raw NIRS signals (i.e., O2Hb and HHb) are contaminated by physiological noise arising from, e.g., heart rate, blood pressure, Mayer waves, and respiration [17]. While—in most studies—either the raw NIRS signals are not used or the physiological noise is filtered out, some studies have investigated the possibility of extracting extra physiological parameters [18,19,20,21].

Respiration is a major source of physiological noise in NIRS that is vividly observable in the spectra of both O2Hb and HHb signals [22]. The emergence of respiration in NIRS measurements is due to (i) the alternation of blood flow within the whole body during the course of inspiration and expiration and (ii) the effect of respiratory fluctuations on cerebral blood volume and flow [23,24]. Some studies have proposed algorithms for separating respiratory components from NIRS measurements, e.g., in [24] by proposing a band-pass filtering (BPF) method. The possibility of RR extraction from NIRS measurements has only been investigated in our previous study [21]. The presented baseline wandering (BW) method, however, was only validated on adult subjects in resting-state measurements which were free from motion artifacts. The presented method was not ideally designed for clinical purposes, i.e., to be implemented in NICU. There are several challenges that exist in data acquisition from hospitalized neonates such as low data quality and patient movements.

Extracting RR from neonatal cerebral NIRS, on the one side, facilitates the concurrent and integrated analysis of breathing and cerebral oxygenation in the NICU. For instance, it could provide a perfectly time-synced analysis of the effect of respiration, including apnea and desaturation events, on cerebral hemodynamics. Deriving RR from neonatal NIRS, on the other side, could potentially reduce the need for adhesive electrodes in the future, which could reduce discomfort, stress, and epidermal stripping, and would promote parent–neonate interaction [25,26,27,28,29,30]. Therefore, the aim of the current study is to develop a novel algorithm, NRR (NIRS RR), for deriving RR from neonatal NIRS measurements recorded in NICUs. With respect to the reference RR recorded with a clinical patient monitor system, we assess the performance of the NRR algorithm in terms of agreement and linear correlation between the reference and extracted RRs. We also compare the NRR algorithm’s performance against the performance of two state-of-the-art methods, i.e., BPF and BW.

## 2. Materials and Methods

### 2.1. Participants

This study was approved by the local ethics committee of the University Medical Center Utrecht (21-098/C). We included ten newborn infants (three females) with gestational age 38 ± 5 weeks and birth weight 3092 ± 990 g that were admitted to the NICU of the Wilhelmina Children’s Hospital, Utrecht, The Netherlands. Among them, three babies were preterm infants (GA < 37 weeks) and one baby was an extremely preterm infant (GA < 28 weeks). They had an indication for clinical NIRS monitoring which was determined by the caring physician or local neuromonitoring protocols. Parents gave informed consent to participate in the study before data acquisition. A total of 19 measurements were recorded from the included neonates without interrupting the clinical routines. After assessing the signal quality as explained in Section 2.3.1, we excluded three measurements due to very low signal quality, leaving 16 measurements from nine neonates (gestational age = 39 ± 3 weeks; weight = 3368 ± 473 g; three females) with a total length of approximately 20.5 h.

### 2.2. Data Acquisition

NIRS signals were recorded at 100 Hz with a cerebral oximetry system (TOM, Artinis Medical Systems B.V., Elst, The Netherlands). The infant-neonatal sensor of the system consists of two transmitters with nominal wavelengths of 760 and 850 nm and a receiver with a 21.5 mm distance from the transmitters, providing an approximate penetration depth of 11 mm, i.e., half the transmitter–receiver distance [31]. The sensor was placed on the neonate’s forehead under a clinical elastic bandage. Figure 1 illustrates the infant-neonatal sensor placed on a baby manikin’s forehead covered by the bandage (Figure 1a) with a schematic of the layout of the transmitters and the receiver, which provides two NIRS channels. The reference RR was recorded with a patient monitor system, Philips IntelliVue MP70 (Philips Medical Systems, Best, The Netherlands), sampled at 0.5 Hz.

### 2.3. Respiratory Rate Extraction Algorithm

The block diagram of the proposed RR extraction algorithm, NRR (NIRS RR), is illustrated in Figure 2. The algorithm comprises two main stages: Stage A, illustrated in Figure 2a; and Stage B, in Figure 2b. Stage A of the NRR algorithm consists of four steps: Preprocessing, HR frequency bandwidth, interquartile range (IQR), and segmentation. These steps are executed once at the beginning of the complete NIRS measurement. Stage B of the NRR algorithm consists of three steps: Motion artifact assessment, HR computation, and RR computation. These are implemented on each 30-second segment of the total hemoglobin concentration (tHb) and IQR signals, computing RR per segment. As respiration influences both O2Hb and HHb signals [22], we have employed the tHb, i.e., the sum of the O2Hb and HHb signals, in order to have a better representation of RR information. We implemented the analyses in this study in Python using *numpy, scipy,* and *matplotlib* modules [32,33].

#### 2.3.1. Preprocessing

Using the modified Beer–Lambert law [34], the optical densities (ODs) of the two NIRS channels are first converted to concentration changes in oxygenated hemoglobin (O2Hb) and deoxygenated hemoglobin (HHb) (OD to Hb in Figure 2a). Then, the signal quality of the NIRS channels is assessed using the Signal Quality Index (SQI) algorithm [35] (signal quality assessment in Figure 2a). The SQI signal is computed in sliding windows of 10 s, overlapping by 50%. Next, the average of the SQI signal across time is computed for each channel, and the channel with the greater average SQI is selected. The whole measurement is excluded in the subsequent analyses if more than 75% of the samples of the SQI of the selected channel are less than an SQI level of 2, which represents low signal quality. Otherwise, the algorithm continues to the subsequent steps.

#### 2.3.2. HR Frequency Bandwidth

First, the tHb signal is low-pass filtered by applying a moving average filter of 1 s and then subtracting it from the original tHb signal. Second, the spectrum of the filtered tHb signal is computed using the multitaper power spectral density estimation [36] based on Slepian sequences [37]. In this study, we used the Python method *scipy.signal.windows.dpss* in order to compute the Slepian sequences by setting the parameters of standardized half bandwidth and the number of windows at 2.5 and 5, respectively. Next, a predefined HR frequency bandwidth between 1.25 Hz and 3.5 Hz is defined according to the HR range of the included patients (75 < HR < 210 beats per minute). This bandwidth is computed once per measurement according to the dominant frequency of the spectrum as follows: the spectrum within the predefined bandwidth is sorted in descending order; the average frequency of the first fifty components (~50% of all components) is then computed; finally, the HR frequency bandwidth ($BW_{HR}$) is set as the average frequency minus 0.5 Hz and plus 0.5 Hz.

#### 2.3.3. Interquartile Range (IQR)

In sliding windows of 1 s, we compute the interquartile range (IQR)—the difference between the third and first quartiles—of the selected tHb signal. Afterwards, the computed IQR values are interpolated to 100 Hz using cubic interpolation. Subsequently, the IQR signal is normalized by dividing it by the median of the tHb signal.

#### 2.3.4. Segmentation

The selected tHb and normalized IQR signals are segmented by sliding windows of 30 s overlapping by 75% (i.e., shifting 7.5 s). Examples of the segmented tHb and IQR are shown in Figure 3a and Figure 3e, respectively. The subsequent steps are implemented on each segmented 30-second tHb and IQR signal, computing RR in each window (i.e., every 7.5 s).

#### 2.3.5. Motion Artifact Assessment

The segmented IQR is thresholded with a threshold of 1%. The thresholded IQR is logically one if the IQR is below the threshold, meaning presumably free from motion artifacts. Figure 3f shows an example of the thresholded IQR signal which has been obtained from the segmented IQR illustrated in Figure 3e. Next, the percentage of samples which are one in the thresholded IQR signal is computed. The segment will be dismissed for RR computation if the percentage is less than 50%. Otherwise, at least 50% of the segment is presumably free from motion artifacts and the algorithm passes to the next steps to compute RR.

#### 2.3.6. HR Computation

The segmented tHb signal is detrended by calculating a least-squares fit of a straight line to the data and then subtracting it (detrending in Figure 2b). Next, the detrended tHb signal is multiplied by the thresholded IQR (multiplication operator in Figure 2b). Figure 3c shows an example of this converted tHb signal, which corresponds to the tHb signal shown in Figure 3a. Then, the spectrum of the converted tHb is computed by using the multitaper power spectral density estimation as explained in Section 2.3.2. Finally, the HR in the considered window is computed by finding the dominant frequency within the $BW_{HR}$. The HR is further used in the next step to limit the allowed RR range.

#### 2.3.7. RR Computation

First, the converted tHb is filtered by using an FIR (finite impulse response) zero-phase band-pass filter implemented with a Kaiser window (band-pass filtering in Figure 2b). The lower and higher cutoff frequencies of the filter are set at 0.1 times the computed HR and 2 Hz, respectively. The higher cutoff frequency was chosen with a 0.5 Hz gap from the maximum reference RR frequency of the patients included (~1.5 Hz). The gap was considered in order to prevent filtering the RR components within the transition bandwidth of the filter. Second, the spectrum of the filtered signal is computed as explained in Section Section 2.3.2 and Section Section 2.3.6. Afterward, an adaptive RR frequency bandwidth ($BW_{RR}$) is defined as 0.15 and 0.85 times the computed HR (i.e., $BW_{RR}=\left[0.15*HR,\;0.85*HR\right]$). Finally, the RR is computed by finding the dominant frequency in the filtered signal within the $BW_{RR}$. Figure 3d illustrates the spectrum of the filtered converted tHb signal corresponding to the converted tHb shown in Figure 3c. The dashed red and green lines and the solid magenta lines depict the reference and extracted RRs and the $BW_{RR}$, respectively. To illustrate the impact of the multiplication by thresholded IQR on the RR computation, Figure 3b shows the spectrum computed from the original tHb signal displayed in Figure 3a, i.e., without any motion artifact assessment and multiplication by the thresholded IQR. It is observed that there is a high error of approximately 0.33 Hz (i.e., 20 breaths per minute, BPM) when the original tHb signal is used, while the error is 0.1 Hz (i.e., 6 BPM) when using the converted signal.

### 2.4. Algorithm Performance Assessment

We employed three quantitative measures for quantifying the agreement between the reference and extracted RRs, including the mean of error (ME) or bias, root mean square error (RMSE), and Bland–Altman limits of agreement (LoA), which is defined as 1.96 times the standard deviation of the error [38]. In addition, as a quantitative measure to determine the linear association between the reference and extracted RRs, we computed Pearson’s correlation coefficient. We assessed the significance of the correlation by performing a Student’s *t*-test with α=1%. Furthermore, we calculated the percentage of the segments wherein RR was computed in each measurement. The mentioned quantitative measures were computed for each measurement separately, and the average and standard deviation for each measure over all measurements were also calculated. Since the reference RR was recorded with 0.5 Hz, each segment of 30 s contains a total of 15 reference RR samples. To have an equal number of samples between the reference and extracted RRs, we have taken the average of the reference RR samples in each 30 s segment.

We compared the performance of the NRR algorithm with two existing methods proposed for extracting RR from NIRS in adults, i.e., band-pass filtering (BPF) [24] and baseline wander (BW) [21]. In these two methods, we set the window length at 30 s, as used in the NRR algorithm. In addition, the cutoff frequencies of the band-pass filter used in the BPF method were set to [0.15, 2] Hz according to the range of the reference RR in this study. To quantitatively compare the performance of the three methods, we computed the average and standard deviation of the mentioned quantitative measures, determining the agreement and linear association between the reference and extracted RRs, over all measurements. As another comparison measure, we defined a boundary wherein the absolute error in RR computation is less than 30% of the mean of the pairwise reference and extracted RRs. We refer to this boundary as the 30% boundary. Furthermore, we performed a Student’s t-test with α=5% to determine whether there is a significant difference in performance between the methods.

### 2.5. Optimization of the Parameters

We conducted a sensitivity analysis of the main parameters used in the NRR algorithm, i.e., the window length and the two constants 0.15 ($A_{low}$) and 0.85 ($A_{high}$) used in defining the adaptive RR frequency bandwidth ($BW_{RR}=\left[A_{low}*HR,\;A_{high}*HR\right]$). Among the 19 recorded measurements, 3 measurements were excluded due to having very low signal quality which was assessed as explained in Section 2.3.1. A total of 20% (i.e., a total of 4.1 h) of the included 16 measurements were randomly selected to optimize the parameters of the NRR algorithm. We computed the RMSE as a measure of error and performed Student’s t-test with α=5% in order to find the statistically optimal parameters.

## 3. Results

### 3.1. Optimization of the Proposed Algorithm’s Parameters

To find a suitable window length, firstly, constants $A_{low}$ and $A_{high}$ were set at 0.05 and 0.95, respectively. Then, different window lengths from 20 to 60 s with steps of 10 s were used. As displayed in Figure 4a, the significantly (*p* < 0.05) lowest RMSE between the reference and estimated RRs was obtained with a window length of 30 s. Different values of $A_{low}$ ranging from 0.05 to 0.2 with steps of 0.05 were investigated to adjust this constant, given the fact that the window length and the $A_{high}$ were set at the optimal window length obtained (30 s) and 0.95, respectively. As shown in Figure 4b, 0.15 and 0.2 are the optimal values for the constant $A_{low}$ which provide significantly lower RMSEs than the other values. However, we selected the lower one (0.15) which makes the bandwidth larger. In the last step, by setting the window length and the constant $A_{low}$ at the optimal ones computed, we varied $A_{high}$ from 0.75 to 0.95 with steps of 0.05 to find the optimal value providing the lowest RMSE. As illustrated in Figure 4c, we found 0.8 and 0.85 as the optimal values for $A_{high}$ providing significantly (*p* < 0.05) lower RMSEs than the other values. However, similarly to the previous step, we selected the higher one (0.85) to set a larger bandwidth for RR.

### 3.2. Results of the Proposed Algorithm

After optimizing the parameters of the algorithm, the remaining 80% (i.e., not included in the sensitivity analysis, see Section 3.1) of the 16 included measurements, a total of 16.4 h or a total of 7872 30-s signal segments, were used for the validation of the proposed algorithm. Figure 5 illustrates the scatter plots of the measurements comparing the reference and extracted RRs. Looking at the reference RR, we observe that the range of reference RR differs between the measurements. As an example, the reference RR in Measurement 2 ranges approximately from 24 to 36 BPM while in Measurement 5 it ranges from approximately 60 to 85 BPM. In addition, it is observed that the patients in some measurements such as Measurements 8, 14, and 15 experienced more variation (~40 BPM) in RR than others. Looking at the scatterplots, we observe that there is a significant linear association (*p* < 0.01) between the reference RR and the extracted RR by the NRR algorithm in all measurements.

Table 1 summarizes the quantitative measures computed in this study to determine the agreement and linear association between the reference and extracted RRs. The ME or bias was lower than approximately 1 BPM (~2% of the average reference RR) in all measurements, except for Measurements 1 (2.4 BPM), 7 (2.7 BPM), and 15, which had the highest bias (8.1 BPM). The average and standard deviation of bias calculated on all measurements are 1.1 and 2.1 BPM (i.e., 1.1 ± 2.1), respectively. The RMSE is lower than 4 BPM in all measurements (exceptions: 1, 4, 7, 15), with the highest one computed for Measurement 15 (12.2 BPM) and average and standard deviation of 3.8 ± 3.0 BPM. The LoA has an average and standard deviation of 6.7 ± 4.7 over all measurements. The highest LoA was obtained in Measurement 15 (18.1 BPM) which had also the highest bias and RMSE. However, looking at Pearson’s correlation, we observe the lowest correlation obtained in Measurement 7 (48%) while it was 71% in Measurement 15. The greatest correlation was obtained in Measurement 5 (96.7%) with an average and standard deviation of 84.5 ± 12.3 computed on all measurements. Looking at the included segments’ percentages, we observe that the RR has been extracted on average for about 94% of the signal segments in all measurements, i.e., leaving on average about 6% of the signal segments with no RR computed due to being highly contaminated by motion artifacts. The lowest percentage of included segments was obtained in Measurement 15 (69.6%). This shows that Measurement 15 was the one in this study with the highest level of contamination by motion artifacts.

### 3.3. Comparison with the Existing Algorithms

Figure 6 shows an example (Measurement 5) of the reference RR and the RR extracted by the proposed NRR algorithm compared with the ones obtained by the BPF and BW methods. It is observed that the BW method works poorly in this measurement in capturing the trend of the reference RR (except for the last ~20 min). The BPF method has provided a better RR extraction than the BW method, providing a Pearson’s correlation of approximately 87%. However, it has some outliers that have errors of more than approximately 20 BPM. On the contrary, the RR extracted by the NRR algorithm has no outliers and follows the trend of the reference RR from the beginning of the measurement to the end, providing a correlation of approximately 97%.

Figure 7 shows the scatterplot and the Bland–Altman plot for the three methods between the reference and extracted RRs when all the measurements are concatenated together. From the scatterplots, it is observed that the RR extracted by the NRR algorithm linearly follows the reference RR in different RR ranges with a Pearson’s correlation coefficient of 95.5% (Figure 7a). This confirms the adaptability of the NRR algorithm to different ranges of RR. Conversely, the scatterplots obtained for the BPF (Figure 7b) and BW (Figure 7c) methods are more sparsely distributed around the identity line than the one obtained for the NRR algorithm. In the scatterplot of the BW method (Figure 7c), the majority of the points are above the identity line, showing the bias of the method to result in a lower RR than the reference RR. Although a significant (*p* < 0.01) correlation was obtained by the BPF and BW methods between the reference RR and the extracted RR, the correlation (60.5% and 34.0%, respectively) is substantially lower than the one obtained by the NRR algorithm. Looking at the Bland–Altman plots, we observe that the BPF and BW methods result in a greater bias in computing RR than the NRR algorithm. Looking at the Bland–Altman limits of agreement (LoA) and comparing it with the 30% boundary (explained in Section 2.4), we observe that the LoA of the NRR algorithm fell within the 30% boundary. This shows that 95% of the error between the reference and extracted RRs, represented by LoA, is lower than 30% of the average RRs. Looking at the Bland–Altman plots of the BPF and BW methods, we observe that the LoA of both methods fell outside the 30% boundary, indicating a higher error, i.e., a wider distribution of error, in RR computation by these methods than the NRR algorithm. The percentage of data points outside of the 30% boundary in the Bland–Altman plots of the NRR, BPF, and BW methods is 2.7%, 22.2%, and 27.4%, respectively. Looking at the bias in the plots, we observe that the BW method has a considerable bias of 7.2 BPM in computing RR, while it is −3.8 and 0.8 BPM for the BPF and NRR methods, respectively.

Table 2 summarizes the quantitative measures determining the agreement and linear association between the reference and extracted RRs by each method, averaged over all measurements. The average of ME or bias computed over all measurements by the NRR (1.1 BPM) and BPF (−1.4 BPM) methods is lower than the one in the BW method (8.4 BPM). However, the standard deviation of ME obtained in the NRR algorithm (2.1 BPM) is much lower than the ones obtained by the BPF (11.6 BPM) and BW (14.1 BPM) methods. The average and standard deviation of RMSE and LoA obtained in the NRR algorithm are 3.8 ± 3.0 BPM and 6.7 ± 4.7 BPM, respectively, which are substantially lower than the ones obtained in the BPF method (RMSE = 10.0 ± 10.7 BPM; LoA = 14.0 ± 11.1 BPM). Although the BW method had a higher average and standard deviation of RMSE (12.6 ± 12.2 BPM) compared with the BPF method, it has a lower average and standard deviation of LoA (11.4 ± 5.8 BPM), which are still higher than the ones obtained in the NRR algorithm. The average and standard deviation of Pearson’s correlation obtained in the NRR algorithm (84.5 ± 12.3%) are greater than the ones obtained by the BPF (70.8 ± 15.0%) and BW (51.1 ± 26.0%) methods. Figure 8 shows the raincloud plot of the absolute error between the reference and extracted RRs for each method. Here, the natural logarithm, i.e., ln(), of the absolute error plus one has been shown for the sake of having a proper visualization of the error. It is observed that the majority (75%, represented by the third quartile in the box plot) of the error in the NRR algorithm fell approximately between 0 and 1, in ln(1+BPM), whereas in the BPF and BW methods the error is more distributed and the majority fell up to approximately 2.5. Looking at the distribution plots, we observe the error was distributed mainly about zero in the NRR algorithm. Conversely, in the distribution plot for the BPF method, there are two additional peaks at about the greatest error, which is due to the outliers as shown in Figure 6. However, the distribution plot of error obtained for the BPF method is less sparse than the one obtained for the BW method, which is roughly uniformly distributed. The result of the statistical comparison between the performance of the three methods in RR extraction against each other showed all three methods performed significantly differently from each other (*p* < 0.05). Therefore, the NRR algorithm performed significantly better than the BPF and BW methods in computing RR.

## 4. Discussion

In this study, we developed a novel algorithm, the NRR algorithm, for extracting RR from neonatal NIRS signals recorded in NICUs. To the best of our knowledge, the proposed NRR algorithm is the first algorithm in the NIRS literature introduced for extracting RR from clinical neonatal NIRS data. The existing methods of RR extraction from NIRS were validated on adult data recorded during the resting stage in a controlled environment (i.e., laboratory environment), whereas the NRR algorithm was validated on neonatal data recorded in a neonatal intensive care unit. We assessed the performance of the NRR algorithm in terms of agreement and linear correlation between the reference and extracted RRs. The results showed a high degree of agreement between the reference and extracted RRs in terms of ME (1.1 ± 2.1 BPM), RMSE (3.8 ± 3.0 BPM), and LoA (6.7 ± 4.7 BPM). In addition, a high linear correlation (r = 84.5 ± 12.3%; *p* < 0.01) was achieved between the reference and extracted RRs.

We compared the performance of the NRR algorithm with two existing methods, i.e., BPF [24] and BW [21]. Compared with the BPF and BW methods, the NRR algorithm showed a stronger agreement between the reference and extracted RRs in terms of ME, RMSE, and LoA. Likewise, the NRR algorithm showed a greater Pearson’s correlation (84.5 ± 12.3%) between the reference and extracted RRs compared with the BPF (70.8 ± 15.0%) and BW (51.1 ± 26.0%) methods. The results of the statistical test implemented confirmed the significant (*p* < 0.05) outperformance of the NRR algorithm over the BPF and BW methods in RR extraction from neonatal NIRS data.

There are several challenges when working with clinical neonatal data compared with healthy adult data. First, neonatal HR and RR have a greater range than the adult ones, and therefore a NIRS system with a higher sampling rate is required. Hence, we used a NIRS system with a 100 Hz sampling rate, whereas the sampling rate was 12.5 Hz in [24] wherein the BPF method was introduced. Second, clinical data, especially the data recorded in the intensive care unit, typically has a lower data quality than the data recorded in a controlled environment due to electromagnetic interference and some limitations in data acquisition which could increase the error in the analyses. Thus, in this study, we used the SQI (Signal Quality Index) algorithm [35] to assess the NIRS signal quality and exclude the signal segments which have very low signal quality, i.e., lower than the SQI level of 2 (low signal quality). Third, clinical data generally has a higher level of motion artifacts due to voluntary and unintentional patient motion than the data recorded in a controlled environment where the subjects’ movements are restricted. In the existing BPF [24] and BW [21] methods, motion artifacts were not considered in the analyses and the signals were recorded during the resting state. Conversely, in this study, we used the IQR (interquartile range) to detect motion artifacts and exclude the signal segments that were highly contaminated with motion artifacts. Fourth, there is more variation in the signal source in clinical recordings than in recordings from healthy subjects in the resting state. Therefore, in this study, we introduced an adaptive RR frequency bandwidth for computing RR that is updated in each segment according to the computed HR as explained in Section 2.3.7. All in all, taking into account the clinical data challenges, we have developed a robust algorithm, NRR, that results in a superior performance than the existing methods in extracting RR from neonatal and clinical NIRS data.

The NIRS system used in this study provides raw NIRS signals (optical densities) with a high sampling rate, i.e., 100 Hz, whereas the existing clinical NIRS systems are incapable of such. High sampling rate NIRS has advantages when extracting physiological information such as RR. One advantage is that it gives the opportunity to assess the signal quality. The NIRS signal quality can be assessed by determining the strength of the heartbeat component in the signals as proposed in [35,39,40], which requires a high sampling rate in order to capture the heartbeats. Another advantage is that, compared with a low sampling rate, it provides a larger number of samples in a specified window; so, with IQR computed in sliding windows of, e.g., 1 s, a better representation of the motion artifacts in the window is achieved.

This study has some limitations that could be addressed in future studies. Due to limited resources and access to participants, especially in an intensive care environment, we were only able to enroll 10 neonates in the study. We chose to enroll 10 neonates based on previous experience with similar studies [19,41,42,43] and the practical constraints of conducting research in a clinical setting. The algorithm proposed in this study was validated on a dataset with a total number of 7872 signal segments. However, a validation of the algorithm is needed on a more extensive dataset acquired from a larger population and a more diverse group of patients in order to recognize this RR monitoring approach as suitable for current clinical routines of RR monitoring in hospitals. The NRR algorithm computes RR every 7.5 s; however, this delay in RR computation could be problematic if analyzing short characteristics of RR is desired. In addition, the NRR algorithm was used as an offline algorithm in this study, but it would be more beneficial in clinical practice to develop an online version of the algorithm. This could be achieved by implementing a few adjustments in Stage A of the algorithm. For instance, the signal quality could be assessed in each 30-s window, and the HR frequency bandwidth could be updated during the measurement every 10 min.

Extracting RR from neonatal NIRS provides the opportunity to have two perfectly synchronized clinical biomarkers—i.e., RR as well as cerebral oxygen saturation—using a single clinical system. RR, on the one side, is known as one of the main vital signs recorded in a standard clinical routine that is an early predictor of clinical deterioration in children [44,45,46,47]. Furthermore, RR monitoring is of great importance in neonates admitted to NICUs as respiratory distress is one of the leading causes of morbidity in the first days of life [48,49,50,51]. NIRS-monitored cerebral oxygen saturation, on the other side, is of great value for determining brain oxygen perfusion, so it has been widely incorporated into the standard clinical routine in NICUs [18]. NIRS monitoring has shown feasibility in diagnosing different clinical conditions such as hypotension, hypoxia, hypocapnia, hypercapnia, anemia, apnea, and asphyxia [10,52,53,54,55,56,57]. Therefore, a NIRS system with the extracted RR opens up the possibility of having a concurrent and complementary assessment of respiration and cerebral perfusion in neonates admitted to the NICU. As a potential use, the system could facilitate the concurrent analysis of RR and cerebral oxygenation for detecting abnormal respiratory events such as tachypnea and apnea and analyzing their effect on cerebral hemodynamics [7,12]. In addition to RR and cerebral oxygen saturation, while applying the NRR algorithm to the raw NIRS signals we could also compute the HR signal (see Section 2.3.6). As a result, we could potentially measure three facets of physiological information, i.e., cerebral oxygen saturation, HR, and RR, using only a single cerebral NIRS system. The portability and easy-to-use features of such a system would make it ideal to be used in the first hours or days of life in clinics, i.e., imposing a minimal burden on the patient and the nurse. In addition, it could prospectively eliminate the need for extra electrodes for RR and HR monitoring with adhesive electrodes. The elimination of excessive and adhesive electrodes would reduce discomfort, stress, and the risk of epidermal stripping, and also could facilitate parent–neonate physical interaction [25,26,27,28,29,58,59].

## 5. Conclusions

In this study, we developed a novel algorithm, NRR (NIRS RR), for extracting RR from clinical NIRS signals recorded in neonates. The results showed a high degree of agreement and a high linear correlation between the reference RR and NIRS-extracted RR. The NRR algorithm outperformed two existing algorithms, i.e., BPF and BW. Therefore, simultaneous RR and cerebral oximetry using a single sensor in a neonatal intensive care setting is feasible. Combining neonatal cerebral NIRS with the extracted RR in a single monitoring system allows for a perfectly time-synced integrated analysis of the impact of abnormal respiratory events (e.g., apnea) on cerebral hemodynamics.

## Figures and Tables

**Figure 1 sensors-23-04487-f001:**
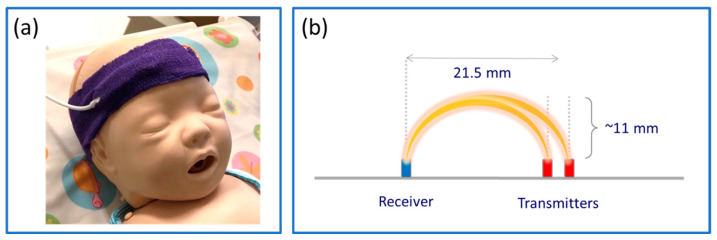
(**a**) A photograph of the infant-neonatal TOM sensor (Artinis Medical Systems B.V., The Netherlands) placed on a baby manikin’s forehead, covered with a clinical self-adhesive elastic bandage. (**b**) Schematic of the configuration of the two transmitters and one receiver embedded in the TOM sensor, illustrated in red and blue, respectively.

**Figure 2 sensors-23-04487-f002:**
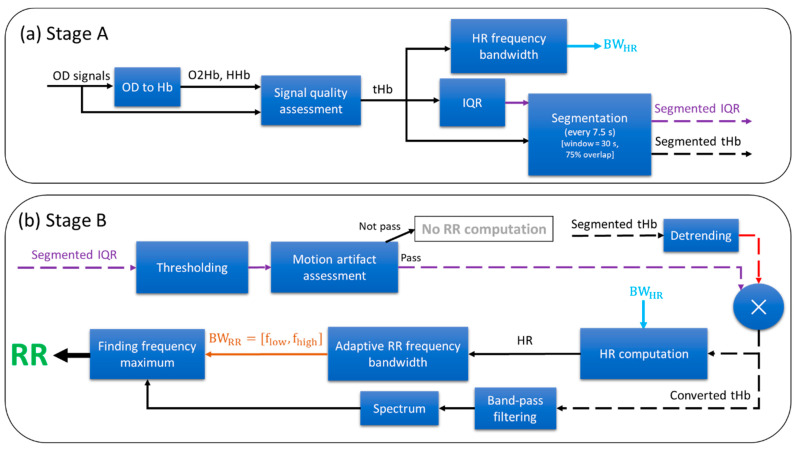
The block diagram of the proposed NRR (NIRS RR) algorithm, comprising Stage A and Stage B. (**a**) Stage A consists of four steps: preprocessing (including OD to Hb and signal quality assessment), HR frequency bandwidth, interquartile range (IQR), and segmentation. (**b**) Stage B consists of three steps: motion artifact assessment (including thresholding and motion artifact assessment), HR computation (including detrending, multiplication operator, and HR computation), and RR computation (including band-pass filtering, spectrum, adaptive RR frequency bandwidth, and finding frequency maximum).

**Figure 3 sensors-23-04487-f003:**
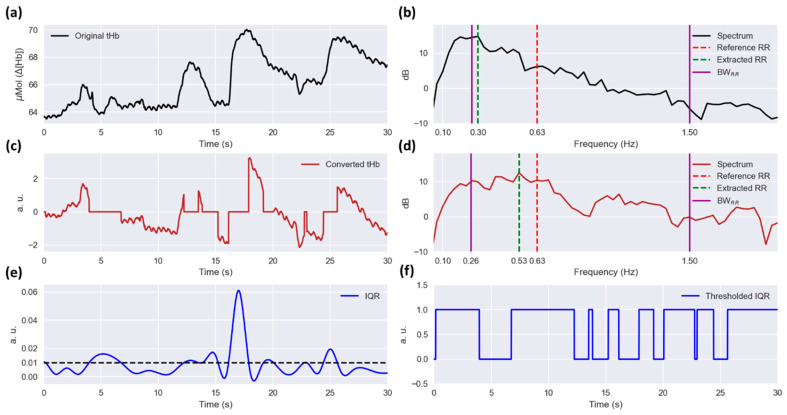
(**a**) A 30-second segment of the tHb signal recorded from one of the subjects (Measurement 15). (**b**) The spectrum computed for the original tHb signal, i.e., without motion artifact assessment and signal conversion. The dashed red and green lines and the solid magenta line depict the reference RR, extracted RR, and adaptive RR frequency bandwidth ($BW_{RR}$), respectively. (**c**) The converted tHb signal after motion artifact assessment and multiplication by the thresholded IQR signal. (**d**) The spectrum of the converted tHb signal. (**e**) The IQR signal computed based on sliding windows of 1 s, interpolated to 100 Hz, and normalized by the median of the tHb signal. The dashed black line depicts the IQR threshold considered for motion artifact assessment (i.e., 1%). (**f**) The thresholded IQR signal. It is zero (i.e., contaminated with motion artifacts) if the IQR is above the defined threshold; otherwise, it is one.

**Figure 4 sensors-23-04487-f004:**
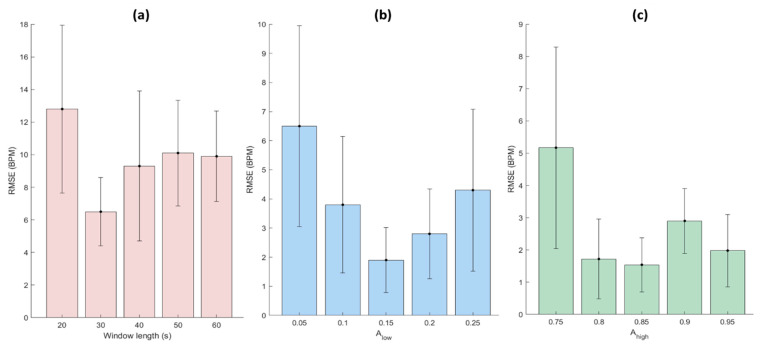
Sensitivity analysis of the main parameters of the NRR algorithm, i.e., window length and constants (i.e., $A_{low},\;A_{high}$) regulating the lower and higher sides of the adaptive RR frequency bandwidth ($BW_{RR}=\left[A_{low}*HR,\;A_{high}*HR\right]$ ). The bar chart of RMSE computed between the reference and extracted RRs with respect to (**a**) the window length, (**b**) the constant $A_{low}$, and (**c**) the constant $A_{high}$ selected, averaged over all measurements. The error bars and bar heights depict the standard deviation and average of RMSE, respectively.

**Figure 5 sensors-23-04487-f005:**
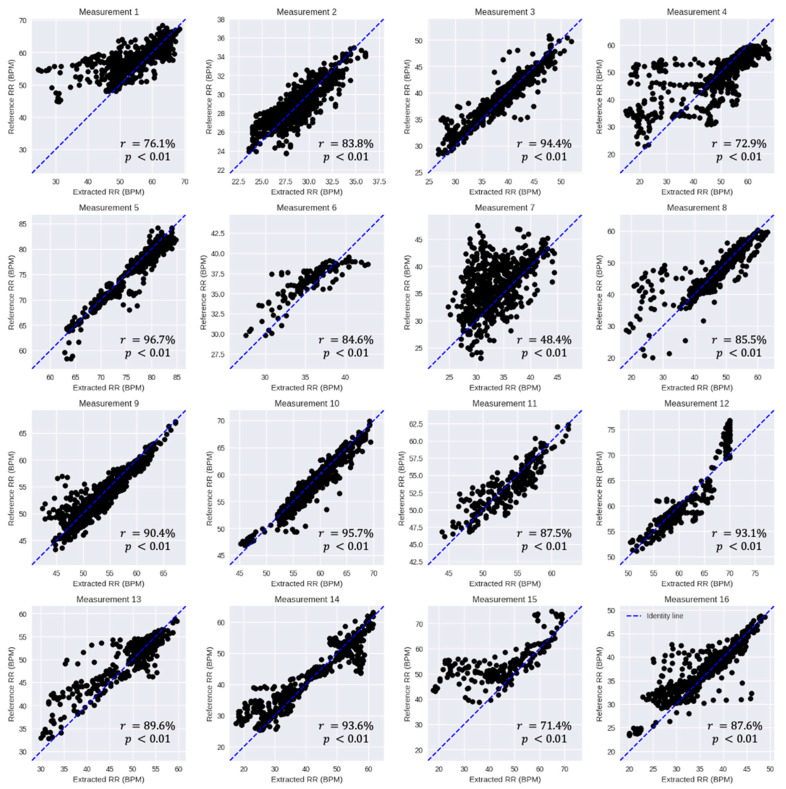
Scatter plots showing the linear association between the reference RR and the extracted RR by the NRR algorithm per measurement. The x and y axes represent the RR in breaths per minute (BPM). Each black dot corresponds to the reference and extracted RRs of each 30-second signal segment. The blue dashed line depicts the identity line (*y* = *x*). The correlation between the reference and extracted RRs is significant in each measurement (*p* < 0.01).

**Figure 6 sensors-23-04487-f006:**
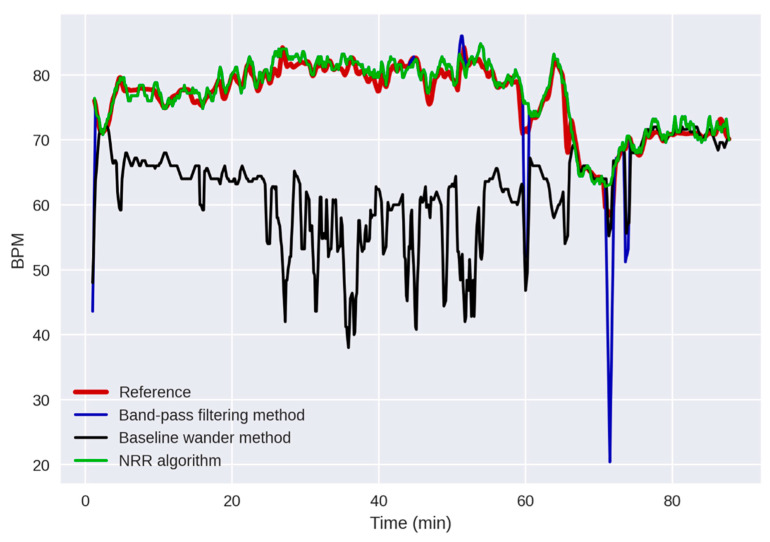
An example of the reference RR signal (in red) and the extracted RR signals obtained by using the NRR algorithm (in green), band-pass filtering (BPF) method (in blue), and baseline wander (BW) method (in black). The x and y axes represent the time and RR in minutes and breaths per minute (BPM), respectively. In this measurement (Measurement 5), Pearson’s correlation magnitude between the reference and extracted RRs is 97% (*p* < 0.01) using the NRR algorithm, 87% (*p* < 0.01) using the BPF method, and 56% (*p* < 0.01) using the BW method, indicating the superior performance of the NRR algorithm than BPF and BW methods.

**Figure 7 sensors-23-04487-f007:**
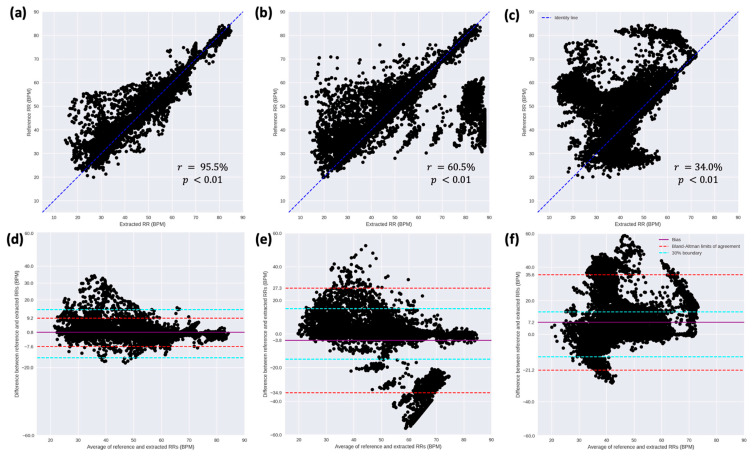
Scatter plots between the reference RR and the extracted RRs using (**a**) the NRR algorithm, (**b**) the band-pass filtering (BPF) method, and (**c**) the baseline wander (BW) method when all measurements are pooled together. Each black dot corresponds to each 30-second signal segment. The Bland–Altman plots of the reference RR and the extracted RRs using (**d**) the NRR algorithm, (**e**) the BPF method, and (**f**) the BW method. The solid magenta line and the dashed red lines depict the bias and the Bland–Altman limits of agreement, respectively. The cyan lines represent the 30% boundary, wherein the error between the reference and extracted RRs is lower than 30% of the mean of the pairwise RRs.

**Figure 8 sensors-23-04487-f008:**
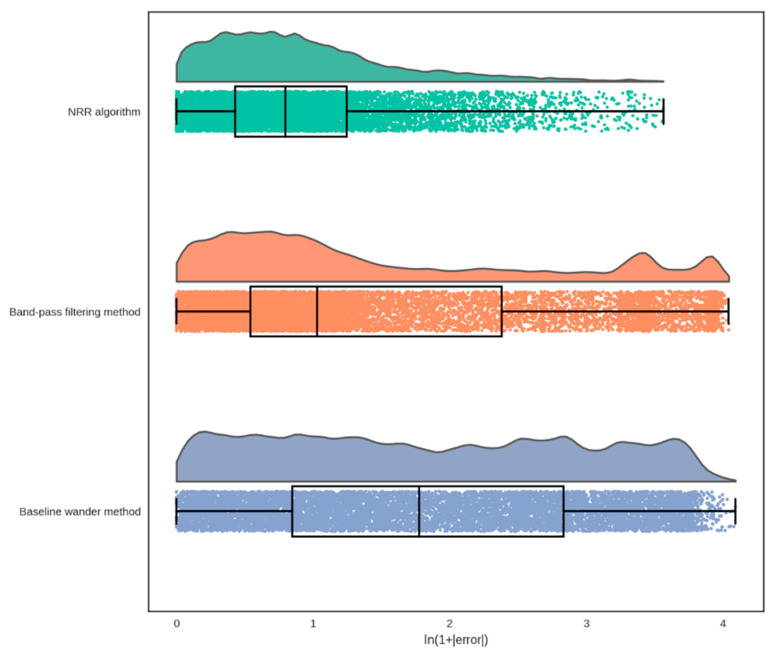
Raincloud plot of the absolute error between the reference RR and the extracted RR by using the NRR algorithm (first panel), the band-pass filtering (BPF) method (second panel), and the baseline wander (BW) method (third panel). The x axis represents the natural logarithm of the absolute error (in BPM) plus one. This was done to have a better visualization of the error distribution.

**Table 1 sensors-23-04487-t001:** Quantitative measures for assessing the performance of the proposed NRR algorithm per measurement.

Measurement	ME ^1^ (BPM ^2^)	RMSE ^3^ (BPM)	LoA ^4^ (BPM)	Pearson’s r (%)	Included Segments (%)
1	2.4	5.9	10.6	76.1	92.1
2	0.2	1.3	2.5	83.8	99.8
3	0.3	1.7	3.2	94.4	99.9
4	0.4	8.1	15.9	72.9	79.2
5	−0.6	1.5	2.6	96.7	99.0
6	0.9	2.0	3.4	84.6	100
7	2.7	5.1	8.4	48.4	99.1
8	0.3	3.8	7.5	85.5	95.5
9	−0.1	2.0	4.0	90.4	96.2
10	−0.5	1.5	2.7	95.7	94.5
11	−0.1	1.8	3.6	87.5	98.8
12	0.0	2.5	4.9	93.1	87.0
13	1.1	3.5	6.5	89.6	96.0
14	0.8	4.4	8.6	93.6	96.5
15	8.1	12.2	18.1	71.4	69.6
16	1.0	2.9	5.7	87.6	97.4
Average	1.1	3.8	6.7	84.5	93.8
Std ^5^	2.1	3.0	4.7	12.3	8.5

^1^ Mean of error. ^2^ Breaths per minute. ^3^ Root mean square error. ^4^ Bland–Altman limits of agreement. ^5^ Standard deviation.

**Table 2 sensors-23-04487-t002:** Quantitative measures for assessing the performance of the proposed NRR algorithm and the two existing methods, i.e., band-pass filtering (BPF) and baseline wander (BW), averaged over all measurements.

Algorithm	ME ^1^ (BPM ^2^)	RMSE ^3^ (BPM)	LoA ^4^ (BPM)	Pearson’s r (%)
NRR	1.1 ± 2.1	3.8 ± 3.0	6.7 ± 4.7	84.5 ± 12.3
BPF	−1.4 ± 11.6	10.0 ± 10.7	14.0 ± 11.1	70.8 ± 15.0
BW	8.4 ± 14.1	12.6 ± 12.2	11.4 ± 5.8	51.1 ± 26.0

^1^ Mean of error. ^2^ Breaths per minute. ^3^ Root mean square error. ^4^ Bland–Altman limits of agreement.

## Data Availability

Not applicable.

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
