# Peer review of "Respiratory Rate Extraction from Neonatal Near-Infrared Spectroscopy Signals"

_sensors, 2023, doi:10.3390/s23094487_

Round 1

Reviewer 1 Report

Dear Authors,

In your paper, you present a novel algorithm for extracting respiration rate from Near Infrared Spectroscopy (NIRS) signals recorded from critically ill neonates. The algorithm involves two steps including Preprocessing, HR frequency bandwidth, Interquartile range, Segmentation, Motion artifact assessment, HR computation, RR computation. The new algorithm’s performance is assessed by comparing the extracted RRs with the reference RRs (measured with a clinical physiological monitor) and with the results of two existing methods. The results confirmed the high correlation between the reference and extracted RR. Moreover, the new algorithm outperformed two existing algorithms.

The treated topic is very interested and the methodology is well described as well as the results; I have just few suggestions to improve the structure of the manuscript, for the benefit of its readability.

 Specific comments

 I suggest to include a specific Section relative to the optimization of the proposed algorithm’s parameters in the Materials and Methods: For example, line 249-251 of the Results could be moved in that specific section. You should better describe the choose of the algorithm’s parameters since this is significant for the performance of your algorithm.

 Figure 7 I suggest to remove lines 370-371 since you already included the information about correlation in the graphs

Regards.

Reviewer 2 Report

The manuscript with the title “Respiratory Rate extraction from neonatal near-infrared spectroscopy signals” is about the extraction of respiratory rate from cerebral NIRS signals in hospitalized neonates using a novel algorithm called NRR (NIRS RR). It deals with a mainly technical topic, but as respiratory rate measurement is an important clinical tool especially in critically ill neonates, it may be an interesting topic for clinicians too. The manuscript is written in appropriate style and manner. There is no major issue to discuss, but there are some minor issues I would like to address.

Generally, the abbreviation list is missing. As it is a technical topic, abbreviation list would be useful for reading.

Abtract:

Clear, well written, conclusions justified

Introduction:

Clear, concise and well written

Materials and Methods:

-          Why have the authors only enrolled term neonates? Is there any reason for that?

-          My concern would be the small convenience number of infants enrolled (only 10 neonates) and at the end only 16 measurement are used to confirm a novel algorithm. There is no need for sample size calculation, as is not feasible in “pilot studies”, however, the actual number of infants included should be predefined and reasoned. Why 10?

Results:

Well written, but there is so many technical information. For me, as clinician it is not completely possible to judge how relevant it is to mention the whole information in the results section, which is very long. It would probably be appropriate to have further opinion of a technically experienced reviewer.

Discussion:

-          There are many NIRS devices in clinical use. The authors have used one of the NIRS device. Would this algorithm work to extract respiratory rate, when another NIRS device is used?

-          According to the authors experience, how long would it take to establish this novel algorithm into a device, so that the parameter respiratory rate can be showed at the display of the NIRS device in form of a number which can be used and interpreted in clinical routine?

-          As it is mentioned in the introduction, that the NIRS signals contain “noise” of physiological processes, is there any chance to extract further information, for example hear rate?

-          The limitation of the study is appropriately acknowledged.

Conclusion:

Well written and clear.

References:

Appropriate
